# Low Self-Perceived Cooking Skills Are Linked to Greater Ultra-Processed Food Consumption Among Adolescents: The EHDLA Study

**DOI:** 10.3390/nu17071168

**Published:** 2025-03-28

**Authors:** Carlos Hermosa-Bosano, José Francisco López-Gil

**Affiliations:** 1Well-Being, Health, and Society Research Group, Universidad de Las Américas, Quito 170513, Ecuador; 2Department of Communication and Education, Universidad Loyola Andalucía, 41704 Seville, Spain; josefranciscolopezgil@gmail.com

**Keywords:** cooking, cooking skills, culinary competence, ultra-processed foods, diet quality

## Abstract

**Introduction:** Ultra-processed foods (UPFs) are widely consumed despite their established associations with obesity, cardiovascular diseases, and other metabolic disorders. One potential factor contributing to high UPF consumption is the decline in cooking skills, particularly among younger generations. This study aimed to describe the cooking skill perceptions of a sample of Spanish adolescents to examine the relationship between perceived cooking skills and UPF consumption, and to identify the specific UPF subcategories most associated with perceived cooking skills. **Methods:** This study is a secondary cross-sectional analysis using data from the Eating Healthy and Daily Life Activities (EHDLA) study, which was conducted among 847 Spanish adolescents (12–17 years) from three secondary schools in *Valle de Ricote* (Region of Murcia, Spain). The participants’ perceptions of their cooking skills were assessed through the following question: “How would you rate your cooking skills?”. The response options included (a) very adequate, (b) adequate, (c) inadequate, and (d) very inadequate. UPF consumption was evaluated via a self-administered food frequency questionnaire (FFQ) previously validated for the Spanish population. UPFs were classified according to the NOVA system, which distinguishes four groups: (1) unprocessed or minimally processed foods; (2) processed culinary ingredients, such as salt, sugar, and oils, used to enhance the preparation of fresh foods; (3) processed foods; and (4) UPF and drink products. To examine the associations between perceived cooking skills and UPF consumption, marginal means and 95% confidence intervals for servings of individual UPF groups were calculated via generalized linear models. These models were adjusted for age, sex, socioeconomic status, physical activity, sedentary behavior, sleep duration, and body mass index to control for potential confounders. Post hoc comparisons between cooking skill categories were conducted via false discovery rate correction following the Benjamini–Hochberg procedure, with significance set at *p* < 0.05. **Results:** Most participants (47%) rated their cooking skills as adequate (47%) or very adequate (18%). Overall UPF intake showed a decreasing trend across skill levels, with the “very adequate” group consuming significantly fewer servings than the “very inadequate” group did (*p* = 0.015). Among the specific UPF categories, adolescents in the “very adequate” category consumed significantly fewer sweets than those in the “very inadequate” and “inadequate” categories did (*p* < 0.05 for all). **Conclusions:** This study revealed evidence of a relationship between cooking skills and overall UPF intake. These results support the importance of interventions that promote cooking competencies among adolescents. School-based culinary programs and community initiatives that teach adolescents simple, time-efficient, and cost-effective cooking techniques could help reduce the overall intake of UPFs.

## 1. Introduction

Ultra-processed foods (UPFs) are industrially modified formulations of food substances that are designed to be hyperpalatable, economic, and convenient [1]. These products are typically high in added sugars, fats, sodium, and artificial additives but low in nutrients such as fiber, protein, vitamins, and minerals [1]. The negative effects of UPFs on diet quality and their link to higher health risks are becoming more apparent [2,3]. Adolescence is a critical period for establishing long-term dietary habits [4], and excessive UPF intake during this stage may predispose individuals to chronic diseases in adulthood. Emerging evidence suggests that early-life exposure to diets high in UPFs is linked to higher risks of obesity [5,6,7], metabolic syndrome [8], cardiovascular diseases [9], type 2 diabetes [10], cancer [11], and other metabolic disorders [12,13]. Additionally, frequent UPF consumption has been associated with cognitive decline [14], all-cause mortality later in life [15], or poorer mental health outcomes in adolescents [16,17]. Given the increasing prevalence of UPFs in global diets, these trends represent a significant public health concern, emphasizing the need for early interventions to promote healthier eating behaviors.

Despite these health risks, UPFs remain widely consumed worldwide because of their affordability, long shelf life, and availability [18,19]. Studies indicate that global UPF consumption has risen significantly, accounting for 20 to 60% of daily energy intake in several middle- and high-income countries [9,12,13,20]. While research on adolescent UPF consumption has primarily focused on Western countries, similar dietary shifts have been observed in low- and middle-income countries, suggesting that these findings may be applicable to a broader range of populations. However, cultural, economic, and policy differences may influence the extent to which cooking skills and dietary habits impact UPF intake in different settings. Despite their widespread consumption, UPFs have not been sufficiently addressed in public health policies. To fill this gap, it is essential to generate strong scientific evidence linking UPF consumption to cognitive and health outcomes, particularly among adolescents. Understanding these associations may help inform targeted interventions aimed at reducing UPF intake and fostering healthier dietary behaviors from an early age.

One potential factor contributing to the widespread reliance on UPFs is the apparent decline in cooking skills, particularly among younger generations [21,22]. Cooking skills encompass a range of abilities and techniques necessary for the planning and preparation of meals with unprocessed, fresh ingredients, such as chopping, mixing, and heating [23,24]. While cooking skills are often evidenced through practical, mechanical tasks, they also encompass perceptual components, including an individual’s confidence, motivation, and willingness to engage in culinary activities [25]. Research has shown that individuals with limited self-perceived cooking abilities are more likely to rely on ready-made meals because of their convenience and ease of preparation [21,26]. Conversely, those with stronger cooking skills tend to make healthier food choices and consume higher amounts of fruits and vegetables [27,28,29]. Furthermore, studies highlight the importance of developing cooking skills from an early age, as proficiency gained during childhood and adolescence has been associated with more frequent meal preparation and lower fast food consumption in adulthood [30].

A recent review by Watanabe and colleagues [22] highlights the limited attention the scientific community has given to the study of cooking skills and their relationship with UPF consumption. Research on this topic has focused primarily on student populations [22], on the basis of the assumption that they frequently eat outside the home and may have lower confidence in their cooking abilities or lack the necessary skills to prepare balanced meals. However, little is known about adolescents’ perceptions of their culinary skills and how these skills are related to their UPF intake. Moreover, most existing studies do not differentiate between UPF subcategories, such as sweetened beverages, confectionery, and dairy-based UPFs, which may have distinct consumption patterns and health implications.

Although the detrimental health effects of UPFs are well documented, there is a gap in understanding how perceived cooking skills influence dietary choices during adolescence. The decline in culinary abilities among younger generations may contribute to the widespread reliance on UPFs, yet this connection has not been sufficiently explored. Given the rising consumption of UPFs and their association with adverse health outcomes, investigating these links could provide valuable insights for designing public health interventions.

Given these knowledge gaps, this study aimed to (1) describe the cooking skill perceptions of a sample of Spanish adolescents, (2) examine the relationship between perceived cooking skills and UPF consumption, and (3) identify the specific UPF subcategories most associated with perceived cooking skills.

## 2. Methods

### 2.1. Study Design and Population

This study is a secondary cross-sectional analysis based on data from the Eating Healthy and Daily Life Activities (EHDLA) study [31]. The sample included Spanish adolescents aged 12–17 years who were enrolled in three secondary schools in *Valle de Ricote*, located in the Region of Murcia, Spain. Data collection was conducted during the 2021–2022 academic year. Of the initial 1378 adolescents, 531 were excluded, resulting in a final sample of 847 participants (45% male).

Participation required written consent from parents or legal guardians, who received an informative document outlining this study’s objectives and surveys. Additionally, the adolescents provided their own consent before taking part in this study. The research protocol adhered to the ethical principles of the Helsinki Declaration and received approval from the Bioethics Committee of the University of Murcia (ID: 2218/2018, 18 February 2019) and the Ethics Committee of the Albacete University Hospital Complex and the Albacete Integrated Care Management (ID: 2021—85, 23 November 2021).

### 2.2. Instruments

Perceived cooking skills: The participants’ perceptions of their cooking skills were assessed through the following question: “How would you rate your cooking skills?”. The response options included (a) very adequate, (b) adequate, (c) inadequate, and (d) very inadequate. This approach has been used in previous research as a quick and efficient method to evaluate perceived culinary competence in adolescents [30]. Although this single-item measure provides a practical and straightforward assessment, it does not capture the full range of cooking skills, and future studies could incorporate more comprehensive instruments to obtain a detailed evaluation.

UPF consumption: Food consumption, as well as energy and nutrient intake, was evaluated via a self-administered food frequency questionnaire (FFQ) previously validated for the Spanish population. The FFQ includes 45 items classified into 12 food groups: (a) red and processed meat; (b) poultry, fish, and eggs; (c) fruits (including preserved fruit); (d) vegetables (salads and other vegetables); (e) dairy products; (f) salted cereals (breakfast cereals, bread, pasta, and rice); (g) sweet cereals (biscuits, pastries); (h) legumes; (i) nuts; (j) sweets (sugars and chocolates); (k) sweetened beverages; and (l) alcoholic drinks. Adolescents reported their food intake on a weekly or monthly basis, allowing for the calculation of average weekly portions for each group. UPFs were classified according to the NOVA system [24], which distinguishes four groups: (1) unprocessed or minimally processed foods; (2) processed culinary ingredients, such as salt, sugar, and oils, used to enhance the preparation of fresh foods; (3) processed foods; and (4) UPF and drink products. The classification of UPFs in this study was conducted using a tailored methodology aligned with previous research. In particular, the UPF categories were defined based on the approach used in the Seguimiento Universidad de Navarra (SUN) cohort study [32]. Additionally, different UPF groups were analyzed separately to offer a more detailed perspective on their potential effects (Appendix A).

Covariates: Sex was self-reported, and age was calculated by asking participants for their birth date. Socioeconomic status (SES) was assessed via the Family Affluence Scale (FAS-III), which includes six items. The total FAS-III score ranges from 0 to 13 points, with higher scores indicating higher socioeconomic status. Body mass index (BMI) was calculated by dividing the participants’ weight in kilograms by their height in meters squared. Overall sleep duration was assessed by asking participants about their usual bedtime and wake-up time on weekdays and weekends. The average sleep duration was calculated via the formula [(weekday sleep duration × 5) + (weekend sleep duration × 2)] divided by 7. The Youth Activity Profile Physical (YAP) questionnaire was used to assess physical activity and sedentary behavior among the adolescents. This self-report questionnaire covered a 7-day period and included 15 items categorized into out-of-school activities, school-related activities, and sedentary habits.

### 2.3. Statistical Analysis

Descriptive statistics were calculated for all study variables, with continuous variables reported as medians and interquartile ranges (IQRs) and categorical variables as absolute frequencies and percentages. Differences in participant characteristics across self-perceived cooking skill levels were evaluated via the Kruskal–Wallis rank sum test for continuous variables and Pearson’s chi-squared test for categorical variables.

To examine the associations between perceived cooking skills and UPF consumption, marginal means and 95% confidence intervals (CIs) for servings of individual UPF groups were calculated via generalized linear models (GLMs) with Gaussian distribution. These models were adjusted for age, sex, socioeconomic status (FAS-III score), physical activity (YAP-S score), sedentary behavior (YAP-S score), sleep duration, and body mass index (BMI) to control for potential confounders. Post hoc comparisons between cooking skill categories were conducted via false discovery rate (FDR) correction following the Benjamini–Hochberg procedure [33]. All the statistical analyses were performed via R software (version 4.4.1). Statistical significance was defined as *p* < 0.05, and all tests were two-tailed.

## 3. Results

Table 1 presents descriptive data of the study participants. In this study, 11% of the participants rated their cooking skills as very inadequate, 25% as inadequate, 47% as adequate, and 18% as very adequate.

Table 2 shows the participants’ data on perceived cooking skills. Significant differences were found for sex, with a greater proportion of male participants reporting inadequate cooking skills and a greater proportion of female participants perceiving their skills as adequate or very adequate (*p* < 0.001). Cooking skills also varied based on socioeconomic status (*p* = 0.004), with higher SES scores among those who rated their skills as very adequate. With respect to lifestyle factors, physical activity scores tended to increase with better cooking skills (*p* = 0.054). Conversely, sedentary behavior scores were significantly lower among those with more adequate cooking skills (*p* = 0.011).

The analysis of estimated marginal means revealed that adolescents with higher self-perceived cooking skills consumed fewer UPFs (Figure 1). Overall UPF intake showed a decreasing trend across skill levels, with the “very adequate” group consuming significantly fewer servings than the “very inadequate” group did (*p* = 0.015).

Among the specific UPF categories, adolescents in the “very adequate” category consumed significantly fewer sweets than those in the “very inadequate” and “inadequate” categories did (*p* < 0.05 for all) (Table 3). Similarly, dairy product intake followed a decreasing non-significant pattern, with the lowest consumption observed among those with very adequate cooking skills. No significant differences in fast food, beverage, or fried food intake were detected across the groups.

## 4. Discussion

This study revealed an association between lower cooking skills and overall UPF consumption in a sample of Spanish adolescents. In addition, this study revealed evidence that lower cooking skills were related to increased consumption of sweets. Our findings align with previous research indicating that adolescents with lower cooking skills tend to consume more UPFs [16,20], reinforcing concerns about the potential long-term health consequences of limited culinary proficiency. Similar studies have shown that individuals with higher culinary competence have greater adherence to healthier diets, including the Mediterranean diet, and lower reliance on processed and convenience foods [23]. However, some studies have suggested that socioeconomic factors and time constraints may mediate this relationship, as even adolescents with adequate cooking skills may still opt for UPFs due to convenience [34]. Adolescence is a key transitional period for the development of dietary habits, and extensive reliance on UPFs may set the stage for poor nutritional patterns that persist into adulthood [20,35]. Therefore, future research should further explore these nuances to clarify the role of additional factors in shaping food choices.

The association between lower cooking skills and increased consumption of UPFs suggests that adolescents with limited culinary confidence may gravitate toward foods that require no preparation or effort. One possible explanation is the greater convenience and accessibility of these products, as well as their deliberate design to be highly palatable and appealing, reinforcing habitual consumption. Furthermore, marketing strategies targeting young consumers may further influence their preference for UPFs over home-cooked meals [36]. Family habits also play a crucial role, as previous research has shown that adolescents whose parents frequently cook at home are less likely to rely on UPFs [34]. Moreover, time constraints can be a determining factor as adolescents often balance school responsibilities, extracurricular activities, and social engagements, reducing the time available for meal preparation and increasing reliance on convenient, ready-to-eat foods [30]. Finally, individuals with lower cooking skills may lack the knowledge, confidence, or interest needed to prepare healthier alternatives, such as homemade snacks or balanced meals [21,35].

Additionally, the home food environment and household cooking frequency may further influence adolescents’ reliance on UPFs. Previous studies have shown that parents and caregivers who lack cooking skills may reinforce the consumption of UPFs among their children [34]. Supporting this notion, recent studies have revealed associations between higher household cooking frequency, lower UPF intake and higher diet quality [37]. Also, peer influence and social norms may shape food choices as adolescents are more likely to consume UPFs when these products are prevalent in school cafeterias, vending machines, and social gatherings. The normalization of UPF consumption within their social circles can reinforce these habits and make healthier alternatives less appealing [38,39].

Psychosocial factors may also contribute to the relationship between cooking skills and UPF consumption. UPFs—especially sweets—are designed to be highly palatable and rewarding, reinforcing habitual consumption patterns [40]. Adolescents experiencing stress, academic pressure, or emotional distress may turn to these foods as a coping mechanism due to their immediate gratification [41]. Additionally, a lack of exposure to nutrition education and limited autonomy in food preparation may reduce adolescents’ motivation to develop healthier eating habits and cooking skills [29]. Future research should explore the psychological and environmental determinants of UPF consumption in greater depth to better inform intervention strategies.

There are several limitations that need to be acknowledged. First, the cross-sectional design of this study does not allow us to establish causal relationships between variables. Moreover, cooking skills were assessed via a single-item measure that asked participants to rate how adequate their cooking skills were. This approach has been previously used and has been proven to be effective as a quick assessment [30]; however, it does not provide enough information regarding adolescents’ actual cooking skills [24]. Several authors have also highlighted the poor conceptualization of what cooking skills are, which aspects matter for healthy eating, and the absence of valid and reliable measures of these skills and behaviors [26,42]. Future research could benefit from employing more comprehensive assessments that include objective measures of cooking proficiency (e.g., standardized cooking tasks or skill demonstrations) alongside self-reported perceptions, especially providing increasing evidence of these factors with healthy eating. In addition, this study is based on secondary data analysis; therefore, the sample was not specifically designed to be representative for this outcome. The original study was representative of overweight/obesity in the *Valle de Ricote*, which should be considered when interpreting the findings.

Despite its limitations, this study has several strengths, such as its focus on adolescents. Understanding this age group is particularly important as it provides valuable insights that can inform interventions aimed at improving culinary competencies [23]. Given the significance of diet quality during adolescence, future research should further explore how cooking skills are related to UPF consumption over time. For example, it would be beneficial to identify which specific cooking skills and behaviors are more strongly related to overall UPF intake, as well as the consumption of specific UPFs. Additionally, determining the minimum set of culinary abilities required to promote healthier food choices, preparation, and intake could help refine intervention strategies. Investigating factors that facilitate or hinder a reduction in UPF intake, alongside other psychosocial aspects such as food literacy, eating context, parental influence, and perceived barriers to cooking, may provide a more comprehensive understanding of the relationship between cooking skills and dietary behaviors [43]. Addressing these factors could contribute to more sustainable and health-conscious dietary patterns among adolescents.

Given these findings, interventions aimed at improving adolescent cooking skills may be an effective strategy to reduce UPF consumption. School-based cooking programs that teach simple, time-efficient, and cost-effective meal preparation techniques have shown promise in improving dietary habits [44]. In addition, parental involvement in meal preparation has been associated with higher diet quality among adolescents. Encouraging parents to cook with their children and promoting home cooking as a family activity could foster greater culinary confidence and healthier food choices [37].

## 5. Conclusions

This study revealed evidence of the relationship between cooking skills and overall UPF intake. These results underscore the need for interventions that promote cooking education among adolescents, emphasizing not only mechanical food preparation skills but also confidence-building and motivation to engage in home cooking. Teaching and promoting culinary skills, particularly among younger generations, offers a promising strategy to reduce reliance on UPFs. By equipping individuals with the knowledge and confidence to cook, professionals can promote healthier and more sustainable eating habits. School-based culinary programs and community initiatives that teach adolescents simple, time-efficient, and cost-effective cooking techniques could help bridge this gap [29,44]. Teaching and promoting culinary skills, particularly among younger generations, offers a promising strategy to reduce reliance on UPFs. By equipping individuals with the knowledge and confidence to cook, professionals can promote healthier and more sustainable eating habits.

## Figures and Tables

**Figure 1 nutrients-17-01168-f001:**
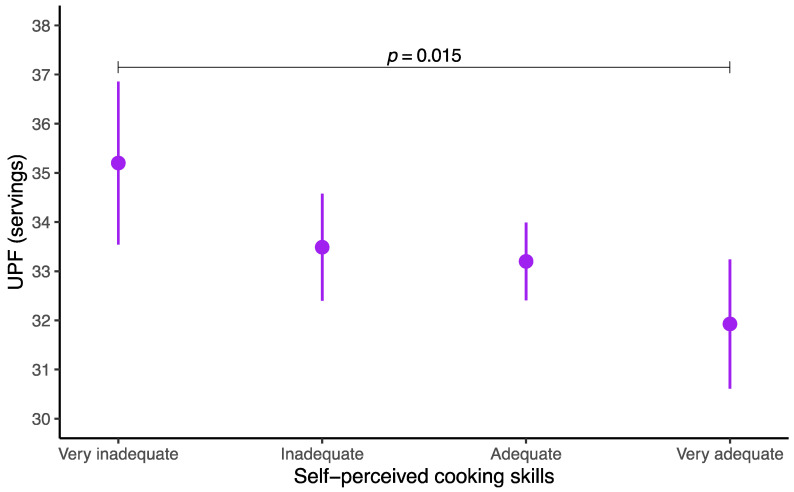
Estimated marginal means of ultra-processed food servings according to self-perceived cooking skills among adolescents. UPF, ultra-processed food. After adjustments for age, sex, socioeconomic status, physical activity, sedentary behavior, sleep duration, and body mass index, corrections for multiple comparisons were applied via the false discovery rate (FDR) *p* value according to the Benjamini–Hochberg procedure [33].

**Table 1 nutrients-17-01168-t001:** Descriptive data of the study participants.

Variable	*N* = 847 ^1^
Age (years)	14.0 (13.0, 15.0)
Sex	
Male	379 (45%)
Female	468 (55%)
FAS-III (score)	8.0 (7.0, 10.0)
Overall sleep duration (minutes)	497.1 (458.6, 527.1)
YAP-S physical activity (score)	2.6 (2.2, 3.0)
YAP-S sedentary behaviors (score)	2.6 (2.2, 3.0)
Energy intake (kcal)	2589.3 (1960.0, 3453.8)
BMI (kg/m^2^)	21.7 (19.3, 25.3)
Sausages (servings)	5.0 (3.0, 7.0)
Fast food (servings)	3.0 (2.0, 5.0)
Dairy products (servings)	3.0 (1.0, 5.0)
Beverages (servings)	3.0 (1.0, 6.0)
Fried foods (servings)	2.0 (1.0, 3.0)
Sweets (servings)	10.0 (5.0, 16.0)
UPF (servings)	26.0 (18.0, 38.0)
Self-perceived cooking skills	
Very inadequate	91 (11%)
Inadequate	210 (25%)
Adequate	396 (47%)
Very adequate	150 (18%)

^1^ Median (interquartile range) or number (%). BMI, body mass index; FAS-III, Family Affluence Scale-III; UPF, ultra-processed food; YAP-S, Spanish Active Profile.

**Table 2 nutrients-17-01168-t002:** Descriptive data of the study participants based on self-perceived cooking skills in adolescents.

Variable	Very Inadequate *n* = 91 ^1^	Inadequate *n* = 210 ^1^	Adequate *n* = 396 ^1^	Very Adequate *n* = 150 ^1^	*p* Value ^2^
Age (years)	14.0 (13.0, 15.0)	14.0 (13.0, 15.0)	14.0 (13.0, 15.0)	14.0 (13.0, 16.0)	0.315
Sex					<0.001
Male	47 (52%)	124 (59%)	160 (40%)	48 (32%)	
Female	44 (48%)	86 (41%)	236 (60%)	102 (68%)	
FAS-III (score)	8.0 (7.0, 9.0)	8.0 (7.0, 9.0)	8.0 (6.5, 9.0)	9.0 (7.0, 10.0)	0.004
Overall sleep duration (minutes)	497.1 (454.3, 518.6)	495.0 (454.3, 522.9)	501.4 (458.6, 529.3)	495.0 (450.0, 527.1)	0.503
YAP-S physical activity (score)	2.5 (2.1, 2.9)	2.6 (2.2, 3.0)	2.6 (2.2, 3.0)	2.7 (2.3, 3.2)	0.054
YAP-S sedentary behaviors (score)	2.6 (2.2, 3.2)	2.6 (2.4, 3.0)	2.4 (2.2, 2.8)	2.4 (2.0, 3.0)	0.011
Energy intake (kcal)	2670.9 (2072.6, 3468.4)	2556.5 (2061.5, 3400.8)	2589.7 (1931.3, 3455.8)	2503.1 (1779.0, 3758.8)	0.793
BMI (kg/m^2^)	22.1 (19.4, 25.1)	21.2 (19.3, 25.3)	21.5 (19.0, 25.1)	22.4 (19.8, 27.0)	0.183
Sausages (servings)	5.0 (3.0, 7.0)	5.0 (3.0, 7.0)	5.0 (3.0, 7.0)	4.0 (2.0, 7.0)	0.250
Fast food (servings)	3.0 (2.0, 5.0)	3.0 (2.0, 5.0)	3.0 (2.0, 5.0)	3.0 (2.0, 5.0)	0.705
Dairy products (servings)	3.0 (1.0, 6.0)	3.0 (1.0, 5.0)	3.0 (1.0, 5.0)	2.0 (0.0, 5.0)	0.032
Beverages (servings)	4.0 (1.0, 9.0)	3.0 (1.0, 6.0)	3.0 (1.0, 6.0)	2.0 (0.0, 6.0)	0.115
Fried foods (servings)	2.0 (1.0, 3.0)	2.0 (1.0, 3.0)	2.0 (1.0, 3.0)	1.0 (0.0, 3.0)	0.144
Sweets (servings)	11.0 (7.0, 16.0)	10.0 (6.0, 16.0)	9.0 (5.0, 16.0)	7.0 (4.0, 14.0)	0.009
UPF (servings)	29.0 (19.0, 39.0)	27.0 (19.0, 39.0)	26.5 (18.0, 38.5)	22.0 (14.0, 37.0)	0.016

^1^ Median (interquartile range) or number (%). ^2^ Kruskal–Wallis rank sum test; Pearson’s chi-squared test. BMI, body mass index; FAS-III, Family Affluence Scale-III; UPF, ultra-processed food; YAP-S, Spanish Active Profile.

**Table 3 nutrients-17-01168-t003:** Estimated marginal means of servings of individual groups of ultra-processed food consumed according to self-perceived cooking skills in adolescents.

Variable	Very Inadequate N = 91 ^1^	Inadequate N = 210 ^1^	Adequate N = 396 ^1^	Very Adequate N = 150 ^1^
Sausages	5.4 (4.9 to 6.0)	5.5 (5.1 to 5.8)	5.2 (5.0 to 5.5)	4.9 (4.4 to 5.3)
Fast food	3.5 (3.1 to 3.8)	3.8 (3.6 to 4.1)	3.9 (3.7 to 4.1)	3.8 (3.5 to 4.1)
Dairy products	4.2 (3.6 to 4.7)	3.6 (3.2 to 3.9)	3.7 (3.4 to 4.0)	3.3 (2.9 to 3.8)
Beverages	4.8 (4.0 to 5.6)	4.1 (3.6 to 4.6)	3.9 (3.6 to 4.3)	4.1 (3.5 to 4.7)
Fried foods	1.9 (1.6 to 2.2)	2.0 (1.8 to 2.2)	2.0 (1.9 to 2.2)	1.8 (1.5 to 2.0)
Sweets	12.4 (11.4 to 13.5)	12.2 (11.5 to 12.9)	12.0 (11.5 to 12.5)	10.7 (9.8 to 11.6) ^a,b,c^
Overall UPF	35.2 (33.5 to 36.9)	33.5 (32.4 to 34.6)	33.2 (32.4 to 34.0)	31.9 (30.6 to 33.2) ^a^

^1^ Data are expressed as estimated marginal means and 95% confidence intervals. Age, sex, socioeconomic status, physical activity, sedentary behavior, sleep duration, and body mass index were adjusted for. ^a^ Significant difference from “very inadequate” (*p* < 0.05). ^b^ Significant difference from “inadequate” (*p* < 0.05). ^c^ Significant difference from “adequate” (*p* < 0.05). UPF, ultra-processed food. Corrections for multiple comparisons were applied utilizing the false discovery rate (FDR) *p* value according to the Benjamini–Hochberg procedure [33].

## Data Availability

The data that support the findings of this study are available from the corresponding author upon reasonable request. The data are not publicly available as they include minors.

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
