# Peer review of "Low Self-Perceived Cooking Skills Are Linked to Greater Ultra-Processed Food Consumption Among Adolescents: The EHDLA Study"

_nutrients, 2025, doi:10.3390/nu17071168_

Round 1

Reviewer 1 Report

Comments and Suggestions for Authors

Before it can be considered for publication, this manuscript needs to be properly revised. These are my suggestions for improvement:

In the abstract, the methods are not clear. Please, specify the applied methodologies. The highlighted results are scarce, much more is expected.

The Introduction is very poor and doesn’t provide the background needed to justify the need to carry out this study. How is this study novel? The expected results are evident.

The number/characteristics of the participants shouldn’t be in Methods. Please, move it to the Results section. How do you know the sample size is adequate/representative of the study population?

The Discussion can be improved and more investigations should be included in this analysis.

The Conclusions should be provided in a separate section.

Author Response

Reviewer 1

Before it can be considered for publication, this manuscript needs to be properly revised. These are my suggestions for improvement:

Thank you for your valuable time and feedback.

 In the abstract, the methods are not clear.

Thank you for your indication. We have modified as follows: “Participants’ perceptions of their cooking skills were assessed through the following question: “How would you rate your cooking skills?”. The response options included (a) very adequate, (b) adequate, (c) inadequate, and (d) very inadequate. UPF consumption was evaluated via a self-administered food frequency questionnaire (FFQ) previously validated for the Spanish population. UPFs were classified according to the NOVA system, which distinguishes four groups: (1) unprocessed or minimally processed foods; (2) processed culinary ingredients, such as salt, sugar, and oils, used to enhance the preparation of fresh foods; (3) processed foods; and (4) UPF and drink products. To examine the associations between perceived cooking skills and UPF consumption, marginal means and 95% confidence intervals for servings of individual UPF groups were calculated via generalized linear models. These models were adjusted for age, sex, socioeconomic status, physical activity, sedentary behavior, sleep duration, and body mass index to control for potential confounders. Post hoc comparisons between cooking skill categories were conducted via false discovery rate correction following the Benjamini‒Hochberg procedure, with significance set at p < 0.05”. 

Please, specify the applied methodologies. The highlighted results are scarce, much more is expected.

We have modified as follows: “Overall UPF intake showed a decreasing trend across skill levels, with the “very adequate” group consuming significantly fewer servings than the “very inadequate” group did (p = 0.015). Among the specific UPF categories, adolescents in the “very adequate” category consumed significantly fewer sweets than those in the “very inadequate” and “inadequate” categories did (p < 0.05 for all)”.

The Introduction is very poor and doesn’t provide the background needed to justify the need to carry out this study. How is this study novel? The expected results are evident.

Thank you for your indication. We have modified as follows:

First paragraph: “Ultra-processed foods (UPFs) are industrially modified formulations of food substances that are designed to be hyperpalatable, economic, and convenient [1]. These products are typically high in added sugars, fats, sodium, and artificial additives but low in nutrients such as fiber, protein, vitamins, and minerals [1]. The negative effects of UPFs on diet quality and their link to higher health risks are becoming more apparent [2,3]. Adolescence is a critical period for establishing long-term dietary habits [4], and excessive UPF intake during this stage may predispose individuals to chronic diseases in adulthood. Emerging evidence suggests that early-life exposure to diets high in UPFs is linked to higher risks risks of obesity [5–7], metabolic syndrome [8], cardiovascular diseases [9], type 2 diabetes [10], cancer [11], and other metabolic disorders [12,13]. Additionally, frequent UPF consumption has been associated with cognitive decline [14], all-cause mortality later in life [15], or poorer mental health outcomes in adolescents [16,17]. Given the increasing prevalence of UPFs in global diets, these trends represent a significant public health concern, emphasizing the need for early interventions to promote healthier eating behaviors.

Second paragraph: “Despite these health risks, UPFs remain widely consumed worldwide because of their affordability, long shelf life, and availability [18,19]. Studies indicate that global UPF consumption has risen significantly, accounting for 20 to 60% of daily energy intake in several middle- and high-income countries [9,12,13,20]. While research on adolescent UPF consumption has primarily focused on Western countries, similar dietary shifts have been observed in low- and middle-income countries, suggesting that these findings may be applicable to a broader range of populations. However, cultural, economic, and policy differences may influence the extent to which cooking skills and dietary habits impact UPF intake in different settings. Despite their widespread consumption, UPFs have not been sufficiently addressed in public health policies. To fill this gap, it is essential to generate strong scientific evidence linking UPF consumption to cognitive and health outcomes, particularly among adolescents. Understanding these associations may help inform targeted interventions aimed at reducing UPF intake and fostering healthier dietary behaviors from an early age”.

Fifth paragraph: “Although the detrimental health effects of UPFs are well-documented, there is a gap in understanding how perceived cooking skills influence dietary choices during adolescence. The decline in culinary abilities among younger generations may contribute to the widespread reliance on UPFs, yet this connection has not been sufficiently explored. Given the rising consumption of UPFs and their association with adverse health outcomes, investigating these links could provide valuable insights for designing public health interventions”.

The number/characteristics of the participants shouldn’t be in Methods. Please, move it to the Results section.

Thank you for your thoughtful feedback. We understand your concern; however, the number and characteristics of the participants are part of the sample description, which is typically included in the Methods section rather than in the Results. This placement ensures clarity and allows readers to understand the study population before interpreting the findings.

How do you know the sample size is adequate/representative of the study population?

Thank you for your comment. This is a secondary study, and the sample is not intended to be representative for this specific outcome. The original study was designed to be representative for overweight/obesity. To address this, we have added the following statement in the limitations section: “This study is based on secondary data analysis; therefore, the sample was not specifically designed to be representative for this outcome. The original study was representative of overweight/obesity, which should be considered when interpreting the findings”.

The Discussion can be improved and more investigations should be included in this analysis.

Thank you for your indication. We have modified discussion section as follows:

First paragraph: “This study revealed an association between lower cooking skills and overall UPF consumption in a sample of Spanish adolescents. In addition, this study revealed evidence that lower cooking skills were related to increased consumption of sweets. Our findings align with previous research indicating that adolescents with lower cooking skills tend to consume more UPFs [16,20], reinforcing concerns about the potential long-term health consequences of limited culinary proficiency. Similar studies have shown that individuals with higher culinary competence have greater adherence to healthier diets, including the Mediterranean diet, and lower reliance on processed and convenience foods [17]. However, some studies have suggested that socioeconomic factors and time constraints may mediate this relationship, as even adolescents with adequate cooking skills may still opt for UPFs due to convenience [28]. Adolescence is a key transitional period for the development of dietary habits, and extensive reliance on UPFs may set the stage for poor nutritional patterns that persist into adulthood [14,29]. Therefore, future research should further explore these nuances to clarify the role of additional factors in shaping food choices”.

Second paragraph: “The association between lower cooking skills and increased consumption of UPFs suggests that adolescents with limited culinary confidence may gravitate toward foods that require no preparation or effort. One possible explanation is the greater convenience and accessibility of these products, as well as their deliberate design to be highly palatable and appealing, reinforcing habitual consumption. Additionally, marketing strategies targeting young consumers may further influence their preference for UPFs over home-cooked meals [30]. Family habits also play a crucial role, as previous research has shown that adolescents whose parents frequently cook at home are less likely to rely on UPFs [28]. Moreover, time constraints can be a determining factor, as adolescents often balance school responsibilities, extracurricular activities, and social engagements, reducing the time available for meal preparation and increasing reliance on convenient, ready-to-eat foods [30]. Finally, individuals with lower cooking skills may lack the knowledge, confidence, or interest needed to prepare healthier alternatives, such as homemade snacks or balanced meals [15,29]”.

Third paragraph: “Psychosocial factors may also contribute to the relationship between cooking skills and UPF consumption. UPFs—especially sweets—are designed to be highly palatable and rewarding, reinforcing habitual consumption patterns [34]. Adolescents experiencing stress, academic pressure, or emotional distress may turn to these foods as a coping mechanism due to their immediate gratification [35]. Additionally, a lack of exposure to nutrition education and limited autonomy in food preparation may reduce adolescents’ motivation to develop healthier eating habits and cooking skills [23]. Future research should explore the psychological and environmental determinants of UPF consumption in greater depth to better inform intervention strategies”.

Sixth paragraph: “Given these findings, interventions aimed at improving adolescent cooking skills may be an effective strategy to reduce UPF consumption. School-based cooking programs that teach simple, time-efficient, and cost-effective meal preparation techniques have shown promise in improving dietary habits [38]. Additionally, parental involvement in meal preparation has been associated with higher diet quality among adolescents. Encouraging parents to cook with their children and promoting home cooking as a family activity could foster greater culinary confidence and healthier food choices [31]”.

The Conclusions should be provided in a separate section.

Done. Thank you.

Reviewer 2 Report

Comments and Suggestions for Authors

This is my review on your manuscript titled "Low Self-Perceived Cooking Skills are Linked to Greater Ultra-Processed Food Consumption among Adolescents: The EHDLA Study" which examines the relationship between adolescents' cooking skills and their intake of ultra-processed foods.

In the introduction you should mention the long-term health risks associated with adolescent UPF consumption and its public health implications and explain how findings might generalize to other populations.

In the materials section you should mention how the single-item cooking skills assessment compares to validated tools and give examples for each category in the NOVA system.

The results are well presented. You could add graphs to emphasize key statistical findings.

Discuss similarities and differences with previous research on adolescent eating behaviors. Why does lower cooking skills lead to increased UPF consumption? What are your recommendations on cooking programs or parental involvement in meal preparation.

Author Response

Reviewer 2

This is my review on your manuscript titled "Low Self-Perceived Cooking Skills are Linked to Greater Ultra-Processed Food Consumption among Adolescents: The EHDLA Study" which examines the relationship between adolescents' cooking skills and their intake of ultra-processed foods.

Thank you four your valuable time and feedback.

In the introduction you should mention the long-term health risks associated with adolescent UPF consumption and its public health implications and explain how findings might generalize to other populations.

Thank you for your indication. We have modified the introduction section as follows: “Ultra-processed foods (UPFs) are industrially modified formulations of food substances that are designed to be hyperpalatable, economic, and convenient [1]. These products are typically high in added sugars, fats, sodium, and artificial additives but low in nutrients such as fiber, protein, vitamins, and minerals [1]. The negative effects of UPFs on diet quality and their link to higher health risks are becoming more apparent [2,3]. Adolescence is a critical period for establishing long-term dietary habits [4], and excessive UPF intake during this stage may predispose individuals to chronic diseases in adulthood. Emerging evidence suggests that early-life exposure to diets high in UPFs is linked to higher risks risks of obesity [5–7], metabolic syndrome [8], cardiovascular diseases [9], type 2 diabetes [10], cancer [11], and other metabolic disorders [12,13]. Additionally, frequent UPF consumption has been associated with cognitive decline [14], all-cause mortality later in life [15], or poorer mental health outcomes in adolescents [16,17]. Given the increasing prevalence of UPFs in global diets, these trends represent a significant public health concern, emphasizing the need for early interventions to promote healthier eating behaviors.

Despite these health risks, UPFs remain widely consumed worldwide because of their affordability, long shelf life, and availability [18,19]. Studies indicate that global UPF consumption has risen significantly, accounting for 20 to 60% of daily energy intake in several middle- and high-income countries [9,12,13,20]. While research on adolescent UPF consumption has primarily focused on Western countries, similar dietary shifts have been observed in low- and middle-income countries, suggesting that these findings may be applicable to a broader range of populations. However, cultural, economic, and policy differences may influence the extent to which cooking skills and dietary habits impact UPF intake in different settings. Despite their widespread consumption, UPFs have not been sufficiently addressed in public health policies. To fill this gap, it is essential to generate strong scientific evidence linking UPF consumption to cognitive and health outcomes, particularly among adolescents. Understanding these associations may help inform targeted interventions aimed at reducing UPF intake and fostering healthier dietary behaviors from an early age”.

In the materials section you should mention how the single-item cooking skills assessment compares to validated tools and give examples for each category in the NOVA system.

Thank you for your indication. Concerning the single-item cooking skills assessment we have modified the procedures section as follows: “This approach has been used in previous research as a quick and efficient method to evaluate perceived culinary competence in adolescents [24]. Although this single-item measure provides a practical and straightforward assessment, it does not capture the full range of cooking skills, and future studies could incorporate more comprehensive instruments to obtain a detailed evaluation”.

Regarding UPF, the next information has been added: “The classification of UPFs in this study was conducted using a tailored methodology aligned with previous research. In particular, the UPF categories were defined based on the approach used in the Seguimiento Universidad de Navarra (SUN) cohort study [26]. Additionally, different UPF groups were analyzed separately to offer a more detailed perspective on their potential effects (Table S1)”. Thank you.

The results are well presented. You could add graphs to emphasize key statistical findings.

Thank you for your valuable suggestions. We appreciate the recommendation to include additional graphs to emphasize key statistical findings. However, we have already included a figure that effectively illustrates our main results, and we prefer to keep the presentation as it is to maintain clarity and conciseness.

Discuss similarities and differences with previous research on adolescent eating behaviors. Why does lower cooking skills lead to increased UPF consumption? What are your recommendations on cooking programs or parental involvement in meal preparation.

Thank you for your indication. We have modified discussion section as follows:

First paragraph: “This study revealed an association between lower cooking skills and overall UPF consumption in a sample of Spanish adolescents. In addition, this study revealed evidence that lower cooking skills were related to increased consumption of sweets. Our findings align with previous research indicating that adolescents with lower cooking skills tend to consume more UPFs [16,20], reinforcing concerns about the potential long-term health consequences of limited culinary proficiency. Similar studies have shown that individuals with higher culinary competence have greater adherence to healthier diets, including the Mediterranean diet, and lower reliance on processed and convenience foods [17]. However, some studies have suggested that socioeconomic factors and time constraints may mediate this relationship, as even adolescents with adequate cooking skills may still opt for UPFs due to convenience [28]. Adolescence is a key transitional period for the development of dietary habits, and extensive reliance on UPFs may set the stage for poor nutritional patterns that persist into adulthood [14,29]. Therefore, future research should further explore these nuances to clarify the role of additional factors in shaping food choices”.

Second paragraph: “The association between lower cooking skills and increased consumption of UPFs suggests that adolescents with limited culinary confidence may gravitate toward foods that require no preparation or effort. One possible explanation is the greater convenience and accessibility of these products, as well as their deliberate design to be highly palatable and appealing, reinforcing habitual consumption. Additionally, marketing strategies targeting young consumers may further influence their preference for UPFs over home-cooked meals [30]. Family habits also play a crucial role, as previous research has shown that adolescents whose parents frequently cook at home are less likely to rely on UPFs [28]. Moreover, time constraints can be a determining factor, as adolescents often balance school responsibilities, extracurricular activities, and social engagements, reducing the time available for meal preparation and increasing reliance on convenient, ready-to-eat foods [30]. Finally, individuals with lower cooking skills may lack the knowledge, confidence, or interest needed to prepare healthier alternatives, such as homemade snacks or balanced meals [15,29]”.

Third paragraph: “Psychosocial factors may also contribute to the relationship between cooking skills and UPF consumption. UPFs—especially sweets—are designed to be highly palatable and rewarding, reinforcing habitual consumption patterns [34]. Adolescents experiencing stress, academic pressure, or emotional distress may turn to these foods as a coping mechanism due to their immediate gratification [35]. Additionally, a lack of exposure to nutrition education and limited autonomy in food preparation may reduce adolescents’ motivation to develop healthier eating habits and cooking skills [23]. Future research should explore the psychological and environmental determinants of UPF consumption in greater depth to better inform intervention strategies”.

Sixth paragraph: “Given these findings, interventions aimed at improving adolescent cooking skills may be an effective strategy to reduce UPF consumption. School-based cooking programs that teach simple, time-efficient, and cost-effective meal preparation techniques have shown promise in improving dietary habits [38]. Additionally, parental involvement in meal preparation has been associated with higher diet quality among adolescents. Encouraging parents to cook with their children and promoting home cooking as a family activity could foster greater culinary confidence and healthier food choices [31]”.

Round 2

Reviewer 1 Report

Comments and Suggestions for Authors

The authors need to address these 2 comments:

The number/characteristics of the participants shouldn’t be in Methods. Please, move it to the Results section.

How do you know the sample size is adequate/representative of the study population?

Author Response

The number/characteristics of the participants shouldn’t be in Methods. Please, move it to the Results section.
Thank you for your comment. We have moved the number and characteristics of the participants to the Results section, as per your suggestion. However, we do not fully agree with this change. These details are descriptive data of the examined sample, not a research finding. Typically, such information is included in the Methods section rather than the Results. We have modified as follows: "Table 1 presents descriptive data of the study participants. The sample consisted of 847 participants (45% boys). The median UPF servings consumed was 26.0 (IQR 18.0 to 38.0). Regarding perceived cooking skills, 11% of the participants rated their cooking skills as very inadequate, 25% as inadequate, 47% as adequate and 18% as very adequate".

How do you know the sample size is adequate/representative of the study population?
Thank you for your indication. The next information has been added in the Methods section: "An a priori power analysis was conducted to determine the minimum sample size required to detect a moderate effect size (R2 = 0.13) in a GLM with 10 predictors (including categorical variables coded as dummy variables), using a standard significance level (alpha [α] = 0.05) and statistical power (1 – beta [β] = 0.80). The analysis, performed using the ‘pwr.f2.test’ function in R, indicated that a sample size of 119 participants would be required to achieve adequate power. Given that the current study includes 847 participants, the statistical power is well above the recommended threshold, ensuring sufficient sensitivity to detect the hypothesized effects". 
Regarding representativeness, the next limitation has been indicated: "In addition, this study is based on secondary data analysis; therefore, the sample was not specifically designed to be representative for this outcome. The original study was representative of overweight/obesity, which should be considered when interpreting the findings".

Round 3

Reviewer 1 Report

Comments and Suggestions for Authors

Thank you for follow my suggestions.